# The Size Effect of Silver Nanoparticles on Reinforcing the Mechanical Properties of Regenerated Fibers

**DOI:** 10.3390/molecules28041750

**Published:** 2023-02-12

**Authors:** Jianjun Guo, Chen Xu, Bo Yang, Hang Li, Guohua Wu

**Affiliations:** 1College of Agriculture, Anshun University, Anshun 561000, China; 2College of Biotechnology and Sericultural Research Institute, Jiangsu University of Science and Technology, Zhenjiang 212100, China; 3College of Environmental and Chemical Engineering, Jiangsu University of Science and Technology, Zhenjiang 212100, China; 4Anhui Province Key Laboratory of Pollutant Sensitive Materials and Environmental Remediation, Huaibei Normal University, Huaibei 235000, China

**Keywords:** regenerated silk fibroin, Ag nanoparticles, wet spinning, mechanical properties, different sizes

## Abstract

Regenerated silk fibroin (RSF), made from discarded silk cocoons, can be processed into regenerated silk fibers by a simple, inexpensive, and environmentally friendly wet-spinning process. However, the breaking strength and toughness of most RSF fibers are lower than those of natural silk. In this study, Ag nanoparticles (NPs) of different sizes were introduced into RSF to form RSF/AgNPs hybrid fibers by wet spinning. The effects of AgNPs of different sizes on the mechanical properties and structure of the hybrid fibers were investigated. The results demonstrated that the mechanical properties of hybrid fibers were significantly improved, especially the breaking strain, after the addition of four different sizes of AgNPs. With the reduction in AgNPs size (2–60 nm), the breaking strength and breaking strain of hybrid fibers tended to increase. The results showed that the hybrid fibers containing 2 nm AgNPs were remarkable, with excellent mechanical properties and toughness, and the breaking strain reached 138.27%, which was far greater than blank RSF fibers (15.02%) and even natural silk (about 21%). The S-FTIR and WAXD showed that, compared with the larger AgNPs, the smaller AgNPs contributed more to the formation of silk fibroin β-sheet and crystallinity, and reduced the β-crystallite size. This study is helpful to understand the relationship between the size of nanoparticles and the mechanical properties of hybrid fibers.

## 1. Introduction

*Bombyx mori* silk fiber has been increasingly applied in the field of medical care, biomedicine, and other new materials due to its unique molecular structure, good mechanical properties, and excellent biocompatibility [1,2,3]. In recent years, researchers have modified and functionalized silk surfaces by various techniques to improve the performance of silk fibers, giving silk a new function and enhance its application value [4,5,6,7]. Wet spinning is a simple, inexpensive, environmentally friendly spinning process; even the discarded silk cocoons can be used to produce regenerated silk fibroin (RSF) solutions, the materials of RSF fibers. However, a drawback of most RSF fibers is that their breaking strength and flexibility are lower than those of natural silk. Therefore, there is an urgent need to find simple and easy methods to improve the mechanical properties of regenerated silk.

It is widely known that the silk fibroin molecule is amphiphilic and can be blended with surface-active nanoparticles such as cellulose [8], carbon nanotubes [9], and monodispersed colloidal particles [10] according to the material design and the hybrid effect between the multiple components. Researchers have modified RSF fibers with nanomaterials that might greatly improve the mechanical properties of RSF, such as carbon nanotubes [9] and graphene oxide [4], and found that the breaking strength of the RSF fibers was greatly improved. For example, Pan et al. [11] found that hydrophilic titanium dioxide nanoparticles had significant strengthening and toughening effects on RSF fibers. As we know, nanoparticles have a size effect and a size reduction may cause changes in macroscopic physical properties [12]. As far as we know, the effect of nanoparticle size on the mechanical properties of RSF fibers has rarely been reported [13]. Therefore, it is necessary to investigate the effect of nanoparticle size effect on the properties of regenerated silk.

We synthesized four different sizes of AgNPs by a one-pot method. RSF/AgNPs hybrid fibers composed of AgNPs of different sizes were prepared by wet spinning. The effects of different sizes of AgNPs on the mechanical properties and the structure of the hybrid fibers were investigated. The mechanical properties of the hybrid fibers were significantly improved after adding different sizes of AgNPs. Besides, S-FTIR and WAXD results demonstrated that the addition of smaller AgNPs was beneficial to the formation of β-sheets and β-crystals, and reduced the size of β-sheet nanocrystals. The study of RSF/AgNPs hybrid fibers with different sizes of AgNPs is helpful to understand the effects of nanoparticle size on the structures and mechanical properties of hybrid fibers. A schematic presentation of our work is given in Figure 1.

## 2. Results and Discussion

### 2.1. Morphology of AgNPs of Different Sizes

The UV–Vis spectra of the AgNPs of different particle sizes are shown in Figure 2. It can be seen that all samples contain typical absorption peaks; as the size of the AgNPs increases, the peaks caused by surface plasmon resonance undergo red shift (Figure 2). When the nanoparticle size is generally less than 3 nm, surface plasmon resonance does not occur [12,14]. We found that the spectrum of 2 nm AgNPs had no obvious absorption peak, which was consistent with the characteristic peak shape for small particle sizes.

The morphology of AgNPs can be observed from their TEM images (Figure 3). As can be seen from Figure 3, AgNPs were spherical and had a small particle size distribution; the average diameters were 2.8 ± 1.5 nm, 19.1 ± 2.4 nm, 37.5 ± 3.1 nm, and 58.1 ± 3.8 nm. They were labeled 2 nm, 20 nm, 40 nm, and 60 nm, respectively.

### 2.2. The Effect of AgNPs on the Mechanical Properties of the Hybrid Fibers

The mechanical properties of the RSF and RSF/AgNPs fibers are shown in Figure 4 and Table 1. The breaking strength and breaking strain of the RSF fibers were 158.13 MPa and 15.02%, respectively (Table 1). Te mechanical properties of RSF/AgNPs fibers were greatly increased compared with pure RSF fibers (Figure 4), indicating that AgNPs provided some enhancement of the mechanical properties of RSF fibers. It was found that the breaking strength and breaking strain of RSF/AgNPs fibers first increased and then decreased with the increase in the AgNPs content with different particle sizes. This is because when the AgNPs content increases above the critical point, it may lead to uneven distribution in RSF fibers.

The results showed that when the AgNPs contents of 2 nm, 20 nm, 40 nm, and 60 nm were 2 wt‰, 0.4 wt‰, 4 wt‰, and 8 wt‰, respectively, the mechanical properties of RSF/AgNPs fibers were outstanding, with breaking strength and breaking strain of 297.97 MPa and 138.27%, 273.74 MPa and 121.61%, 242.61 MPa and 93.41%, 200.29 MPa and 83.46%, respectively (Table 1). The breaking strain of hybrid fibers containing AgNPs of different particle size was superior to RSF fibers and even natural silk. Table 1 shows the decrease in breaking strength and breaking strain of RSF/AgNPs fibers with the increase in the size of AgNPs. The breaking strain of RSF fibers with 2 nm AgNPs was the strongest among these hybrid silks, and their toughness was also obviously improved. The breaking strain was greatly increased from 15.02% to 138.27%, an increase of 8.21 times, which was superior to the reported RSF fibers, including RSF fibers made by different methods and even natural silk [4,11,15,16,17].

Obviously, smaller AgNPs made a greater contribution to the mechanical properties of RSF hybrid fibers. The reasons for this may be multifaceted, including smaller nanoparticles having a wider distribution and larger specific surface area [18]. The main interactions between silk fibroin and nanomaterials include intermolecular hydrophobic interactions, electrostatic interactions, π–π interactions, hydrogen bonding interactions, van der Waals and salt bridge interactions [19]. Disulfide bonds are mainly used to connect peptides in silk fibroin and can be broken to form a sulfhydryl group (-SH) by a formic acid dissolution system. The disulfide bonds and salt bridges that maintain the protein’s internal stability show excellent resistance to deformation and recovery [20]. We think that AgNPs may interact with sulfhydryl groups of silk fibroin to form coordination bonds, allowing silk fibroin to tightly connect AgNPs. The chemical bond force is stronger than the intermolecular hydrophobic interaction, electrostatic force interaction, π–π interaction, hydrogen bonding interaction, van der Waals, and salt bridge interaction, meaning that the RSF/AgNPs hybrid fibers have better mechanical properties than the blank RSF fibers, which affects their secondary structure and crystalline structure.

### 2.3. The Effect of AgNPs on the Morphology of the Hybrid Fibers

The SF solution is extruded through a needle into the coagulation bath, where it then precipitates to form RSF fibers. RSF fibers prepared by wet spinning generally have a sheath–core structure. Generally, this structure is relatively loose and has poor mechanical properties. Stretching RSF can not only reduce the sheath–core structure, but also align the molecular chains along the fiber, which can improve the mechanical properties of RSF fibers. The diameter of hybrid fibers in this study is about 32–33 μm (Figure 5). Their surface is a smooth structure (Figure 6) and their internal structure is dense (Figure 7). Compared with RSF fibers, the addition of AgNPs had no significant effect on the morphology of RSF/AgNPs hybrid fibers. This was the basis for maintaining the excellent mechanical properties of RSF/AgNPs hybrid fibers.

### 2.4. The Effect of AgNPs on Secondary Structure and Crystalline Structure of the Hybrid RSF Fibers


**SR-FTIR analysis**


Figure 8 shows the FTIR spectrum of the RSF and RSF/AgNPs fibers, containing four significant peaks at 1232 cm^−1^ (amide III, attributed to random coil or helix or both), 1265 cm^−1^ (amide III, attributed to β-sheet), 1644 cm^−1^ (amide I, attributed to random coil or helix or both), and 1691 cm^−1^ (amide I, attributed to β-turn) are found [21], confirming the co-existence of random coil/helix, β-turn, and β-sheet conformations. It was observed that the FTIR spectra were not significantly different among the RSF/AgNPs fibers, indicating that AgNPs had no significant effect on the main structure of RSF fibers.

In this study, the secondary structure content distribution of RSF and RSF/AgNPs fibers was determined by deconvolution of the amide III region (1200~1300 cm^−1^) [21]. The distribution of the secondary structure content of RSF/AgNPs fibers are shown in Figure 9. It was observed that all RSF/AgNPs fibers had more β-sheet and less random coil/α-helix than RSF fibers. It can also be found that after adding AgNPs of different sizes, the corresponding β-sheet content in RSF/AgNPs fibers decreases as the size of AgNPs increases, and the mechanical properties of hybrid fibers also decrease.

Nanomaterials have very small dimensions and can provide a large number of surface/interface structures. These interfaces can effectively hinder the dislocation slip movement to increase the strength of the material. In addition, nanomaterials have a size effect; a smaller structural unit may make the deformation more uniform, improving the plasticity of the materials. The size of the nanomaterial also affects the aggregation form of some proteins. For example, GO has a size effect on the aggregation of amyloid peptide Aβ (the amyloid β-protein) [22], Aβ has fewer β-sheets and amyloid fibers as the size of GO increases. In this study, all the RSF/AgNPs fibers also had more β-sheets and fewer random coil/α-helix with smaller AgNPs. The interaction between AgNPs and silk fibroin might help the transition of silk fibroin conformation from random coil/α-helix conformation to β-sheet.


**Wide-angle X-ray diffraction (WAXD) analysis**


WAXD is another technology for studying the general structure of silk fibers. The crystallinity and particle size of silk can be determined by WAXD [23,24,25]. Figure 10 shows the WAXD pattern of RSF fibers and AgNPs/RSF fibers with the best mechanical properties with different sizes of AgNPs. It can be observed that the WAXD patterns do not show any obvious difference between the fibers, indicating that the crystalline structure of fibers are not changed by the AgNPs, which is consistent with the results of SR-FTIR. The deconvolution results of RSF and RSF/AgNPs fibers are shown in Figure 10 and Table 2. The mean crystallite size of β-sheet in the a direction (inter-chain), b direction (inter-sheet), and c direction (along the fiber axis) was obtained by Scherrer formula [26,27]. The results show that the AgNPs/RSF fibers have smaller β-sheet nanocrystal size in a, c directiosn and higher crystallinity than RSF fibers. Importantly, with the decrease in AgNPs particle size, the crystallinity of AgNPs/RSF hybrid fibers increased and the β-sheet nanocrystal size in the hybrid fibers decreased. Obviously, this structural change may significantly affect the properties of the fibers [28].

It is reported that some monodisperse nanoparticles can provide nucleation centers and silk fibroin molecules can crystallize on the surface of them [10,29]. Smaller AgNPs could provide more surface area, which may promote the heterogeneous nucleation of silk fibroin molecules on/near the AgNPs surface, and result in more nucleation centers, thus forming higher crystallinity.

## 3. Materials and Methods

### 3.1. Preparation of Silver Nanoparticles (AgNPs)

Synthesis of 2 nm AgNPs: The synthesis method of AgNPs is based on reference [30]. Briefly, AgNO_3_ solution (2 mL, 100 mM) and GSH solution (1 mL, 200 mM) were added into 97 mL DI water. The pH of the solution was then tuned by adding 80 μL 5 M NaOH solution to obtain a clear solution. Subsequently, 200 μL freshly prepared NaBH_4_ solution (0.5 M) was quickly added into the above mixture under vigorous stirring (>900 rpm) for 2 days at R.T.

Synthesis of 20, 40, 60 nm AgNPs: The AgNPs were synthesized with reference to [31]. Sodium citrate (SC) (5 mM) and tannic acid (TA) were added into 100 mL DI water and the mixture was stirred and heated to boiling. After that, AgNO_3_ (1 mL, 25 mM) was injected into this solution under vigorously stirring. The size of the AgNPs was controlled by adjusting the concentrations of TA, which were 0.1 mM, 1 mM, and 5 mM, respectively. The above reagents were purchased from Shanghai Aladdin Biochemical Technology Co., Ltd. (Shanghai, China).

### 3.2. Preparation of RSF Fibers

Silk degumming: The silkworm cocoons (Chinese Academy of Agricultural Sciences, Zhenjiang, China) were boiled in Na_2_CO_3_ aqueous solution (0.05% *w*/*v*) at a bath ratio of 1:20 for 30 min, and then rinsed three times with 60 °C DI; this operation was repeated three times and the degummed silk was then dried in an oven at 45 °C.

Preparation of RSF solution: The degummed silk was dissolved in a 5% (*w*/*v*) CaCl_2_-FA (formic acid) solution for 4 h to prepare a 15 wt% (mass fraction) silk fibroin (SF) solution for wet spinning. AgNPs of different masses and sizes were added into the SF solution. Here, the AgNPs concentration (c) refers to the ratio of AgNPs to SF mass. In this study, the concentration of AgNPs (c) in the SF solutions were 0.2 wt‰, 0.4 wt‰, 0.8 wt‰, 2 wt‰, 4 wt‰, 8 wt‰, 10 wt‰, 14 wt‰, which were named as RSF/AgNPs-0.2, RSF/AgNPs-0.4, RSF/AgNPs-0.8, RSF/AgNPs-2, RSF/AgNPs-4, RSF/AgNPs-8, RSF/AgNPs-10, and RSF/AgNPs-14, respectively.

Wet spinning: All experiments were performed at 24 °C. The spinning solution was poured into a medical syringe and air bubbles were removed by static placement. The spinning solution in the syringe was squeezed vertically into the coagulation bath by a high-pressure injection pump, where the spinning solution was rapidly condensed into uniform fibers. The inner diameter of the spinning needle (26 G) was 0.23 mm. After stretching treatment, the RSF fibers were placed in 75% ethanol solution for 2 h to remove the residual solvent within the fibers. Finally, the RSF fibers were taken out and dried at 24 °C.

### 3.3. Characterization of Silver Nanoparticles and Regenerated Silk

The ultraviolet–visible (UV–Vis) spectra of AgNPs were measured with a UV–Vis spectrophotometer (UV1000F, Labsphere, Halma, NH, USA). The morphology of AgNPs was investigated by transmission electron microscopy (TEM, Tecnai 12, Philips, Amsterdam, The Netherlands). The diameter and morphology of the RSF fibers were characterized via a scanning electron microscope (SEM, Quanta 400 FEG, Reston, VA, USA). The mechanical testing of the RSF fibers was performed by a universal testing machine (Instron 3343, Norwood, MA, USA) in a constant-temperature and -humidity room (24 °C, 65% RH), with a gauge length of 6 mm and a loading rate of 2 mm·min^−1^. At least 20 fibers (6 mm) from the same sample were selected for mechanical testing, and the stress–strain data closest to the average value of 20 fibers (6 mm) was the final representative data of the sample. The structure of RSF/AgNPs hybrid fibers were analyzed by wide-angle X-ray diffraction (WAXD) (Rigaku TTR-III, Tokyo, Japan). Fourier transform infrared (FTIR) spectra of RSF/AgNPs hybrid fibers were characterized by the Shanghai Synchrotron Radiation Facility (SSRF).

## 4. Conclusions

In this study, we prepared RSF/AgNPs hybrid fibers containing AgNPs of different sizes by wet spinning, and studied the effects of different sizes of AgNPs on the mechanical properties and structure of the hybrid fibers. The results demonstrated that the mechanical properties of the hybrid fibers with four different sizes of AgNPs were greatly improved, especially the breaking strain. As the increase size of the AgNPs increased, the breaking strength and breaking strain of the hybrid fibers followed a downward trend. The hybrid fibers with 2 nm AgNPs were prominent due to their remarkable mechanical properties and toughness; the breaking strain reached 138.27%, which was far greater than the blank RSF fibers and even natural silk. The SEM, S-FTIR, and WAXD analyses showed that, compared with the RSF fibers, the morphology and structure of the RSF/AgNPs hybrid fibers were well preserved with more β-sheets and greater crystallinity, as well as smaller crystallite size, which could lead to the increase in mechanical properties of RSF/AgNPs hybrid fibers. 

We believe the AgNPs could help the formation of silk fibroin β-sheets, reducing the silk fibroin β-sheet nanocrystal size. It was considered that AgNPs could interact with silk fibroin to form coordination bonds, which could affect the secondary structure and crystalline structure of hybrid fibers, conferring better mechanical properties to the RSF/AgNPs hybrid fibers than to the RSF fibers. Moreover, this ability to improve the mechanical properties of RSF/AgNPs hybrid fibers by changing the mesostructure of silk fibroin was enhanced as the size of AgNPs becomes smaller. Smaller AgNPs could significantly improve the mechanical properties, especially toughness. This study describes an effective way to improve the properties of RSF fibers from the perspective of size effect, and will have very broad application prospects in the field of material science.

## Figures and Tables

**Figure 1 molecules-28-01750-f001:**
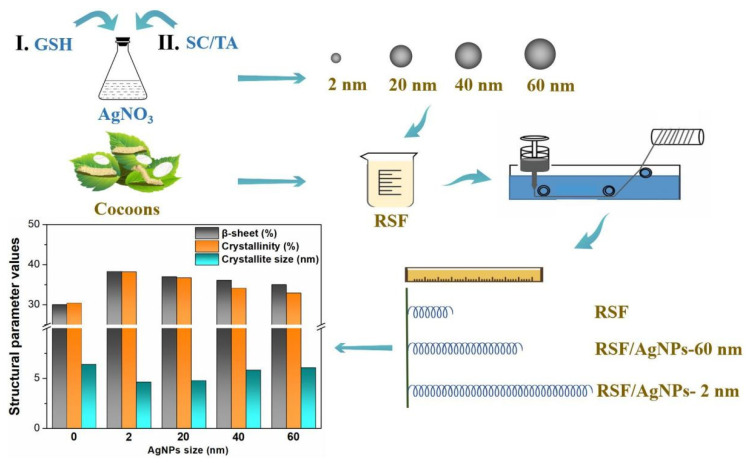
Schematic illustration of the size effect of silver nanoparticles on the reinforcement of the mechanical properties of regenerated fibers.

**Figure 2 molecules-28-01750-f002:**
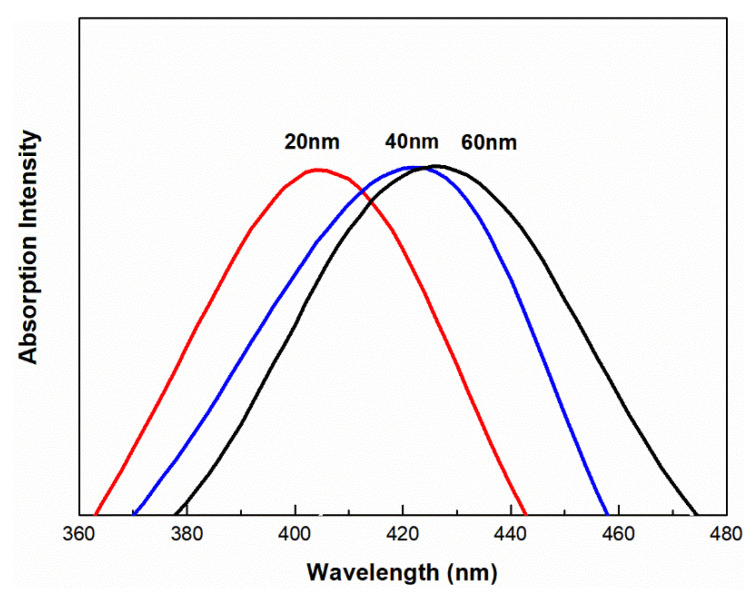
UV–Vis spectrum of 20 nm, 40 nm, 60 nm AgNPs.

**Figure 3 molecules-28-01750-f003:**
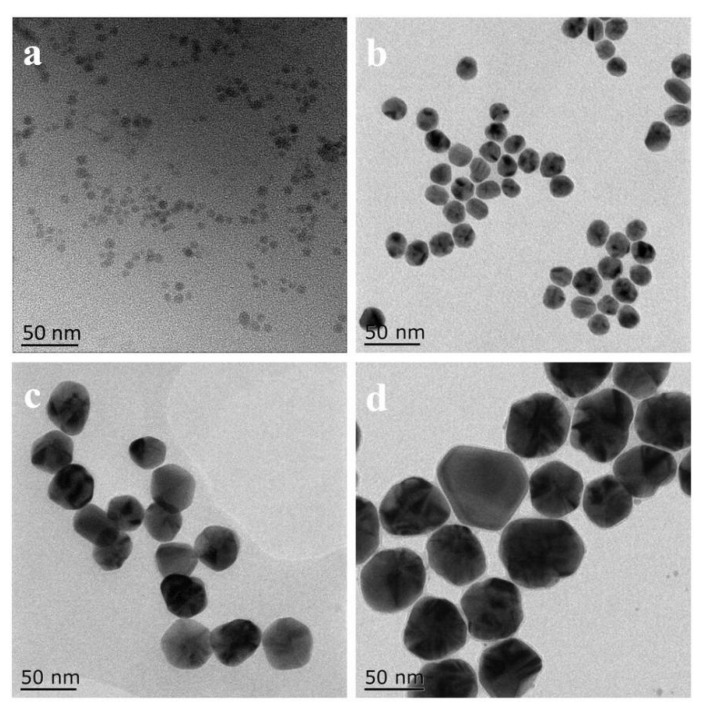
TEM images of (**a**) 2 nm, (**b**) 20 nm, (**c**) 40 nm, (**d**) 60 nm AgNPs.

**Figure 4 molecules-28-01750-f004:**
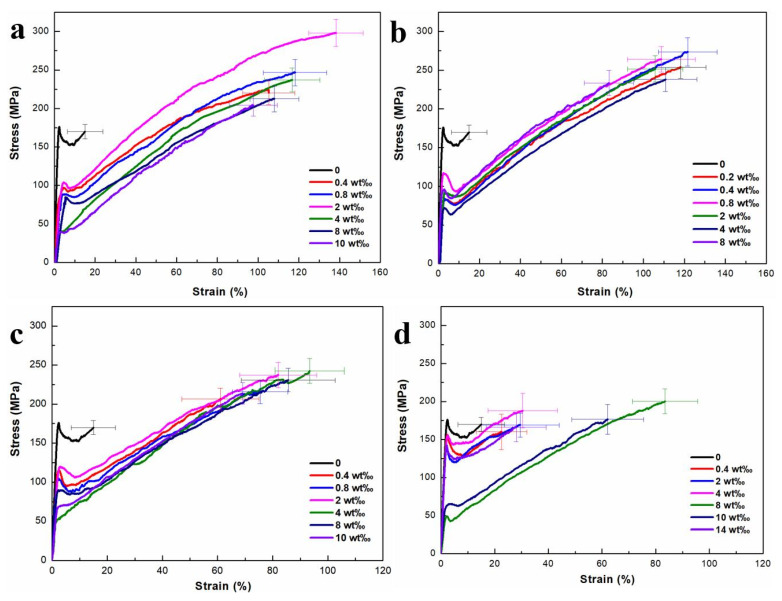
Stress–strain curves of hybrid fibers of RSF/AgNPs (**a**) 2 nm, (**b**) 20 nm, (**c**) 40 nm, (**d**) 60 nm.

**Figure 5 molecules-28-01750-f005:**
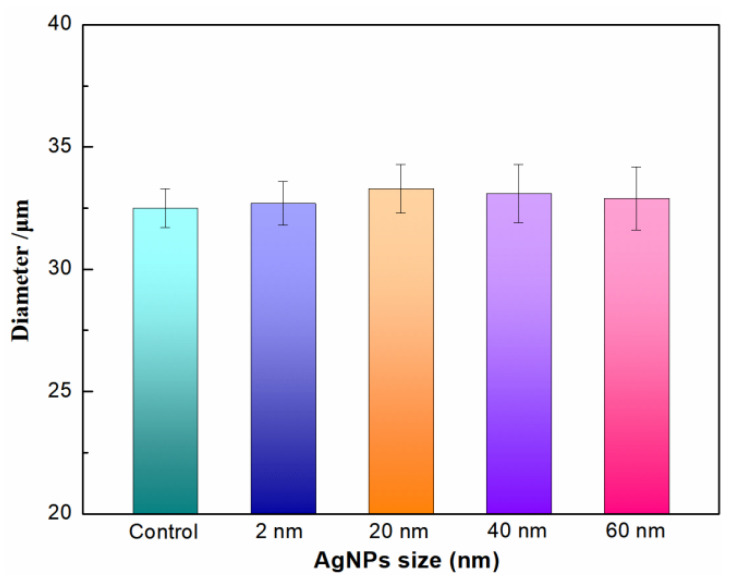
The diameter of RSF fibers and RSF/AgNPs fibers with different sizes of AgNPs.

**Figure 6 molecules-28-01750-f006:**
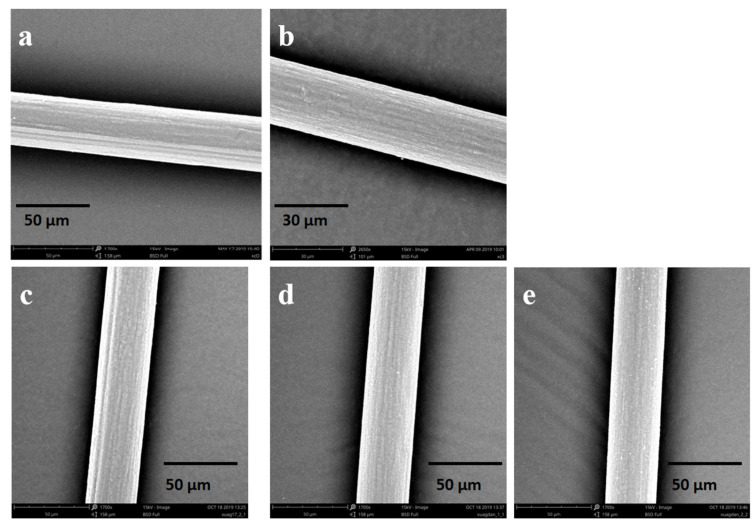
SEM images of surface structure of the (**a**) RSF fibers and RSF/AgNPs fibers with (**b**) 2 nm-2 wt‰, (**c**) 20 nm-0.4 wt‰, (**d**) 40 nm-4 wt‰, (**e**) 60 nm-8 wt‰.

**Figure 7 molecules-28-01750-f007:**
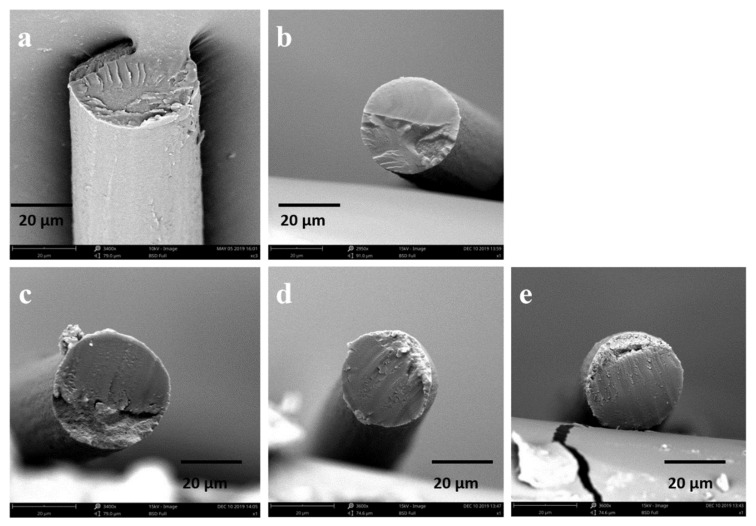
SEM images of internal structure of (**a**) RSF fibers and RSF/AgNPs fibers with (**b**) 2 nm-2 wt‰, (**c**) 20 nm-0.4 wt‰, (**d**) 40 nm-4 wt‰, (**e**) 60 nm-8 wt‰.

**Figure 8 molecules-28-01750-f008:**
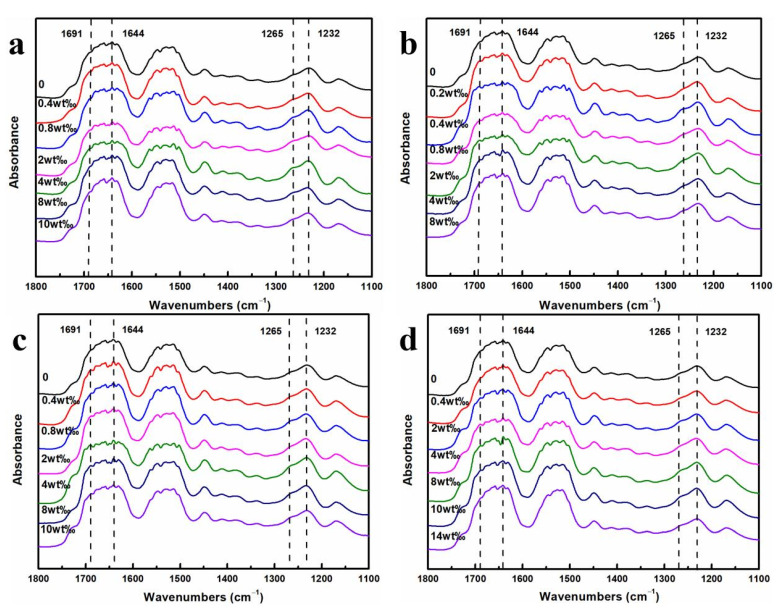
FTIR spectra of RSF/AgNPs hybrid fibers: (**a**) RSF/2 nm AgNPs, (**b**) RSF/20 nm AgNPs, (**c**) RSF/40 nm AgNPs, (**d**) RSF/60 nm AgNPs.

**Figure 9 molecules-28-01750-f009:**
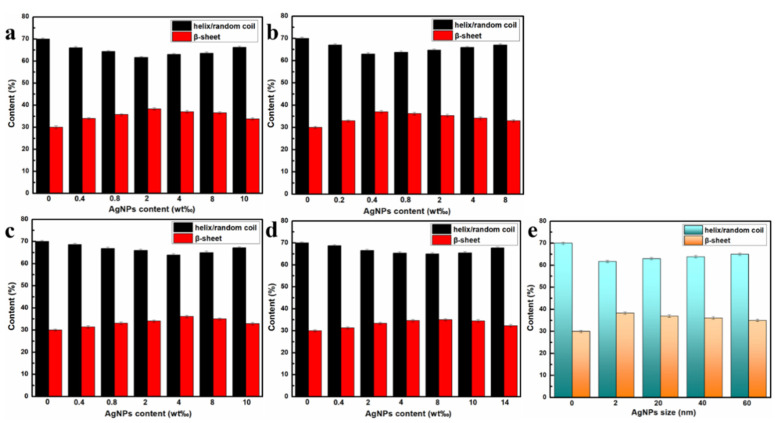
The distribution of the secondary structure content of RSF/AgNPs hybrid fibers: (**a**) RSF/2 nm AgNPs, (**b**) RSF/20 nm AgNPs, (**c**) RSF/40 nm AgNPs, (**d**) RSF/60 nm AgNPs, (**e**) the RSF fibers and RSF/AgNPs fibers with 2 nm-2 wt‰, 20 nm-0.4 wt‰, 40 nm-4 wt‰, 60 nm-8 wt‰.

**Figure 10 molecules-28-01750-f010:**
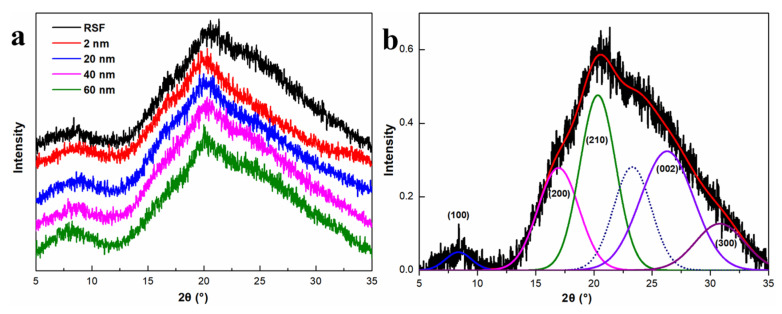
(**a**) WAXD pattern of RSF fibers and RSF/AgNPs hybrid fibers with AgNPs of different size; (**b**) WAXD pattern deconvolution of RSF fibers.

**Table 1 molecules-28-01750-t001:** The mechanical properties of RSF/AgNPs fibers (different sizes).

Sample	Breaking Strength/MPa	Breaking Strain/%
RSF	158.13 ± 9.27	15.02 ± 8.67
RSF/AgNPs-2 nm-2 wt‰	297.97 ± 17.51	138.27 ± 13.29
RSF/AgNPs-20 nm-0.4 wt‰	273.74 ± 18.36	121.61 ± 14.32
RSF/AgNPs-40 nm-4 wt‰	242.61 ± 15.91	93.41 ± 12.58
RSF/AgNPs-60 nm-8 wt‰	200.29 ± 16.28	83.46 ± 12.17

**Table 2 molecules-28-01750-t002:** The crystallinity and crystallite size of RSF/AgNPs hybrid fibers.

Samples	Crystallite Size (nm)				
	a	b	c	V/nm^3^	Crystallinity
RSF	1.92	2.47	1.35	6.40	30.41%
RSF/AgNPs-2 nm-2 wt‰	1.65	2.35	1.20	4.65	38.21%
RSF/AgNPs-20 nm-0.4 wt‰	1.67	2.33	1.23	4.79	36.74%
RSF/AgNPs-40 nm-4 wt‰	1.83	2.57	1.24	5.83	34.12%
RSF/AgNPs-60 nm-8 wt‰	1.87	2.56	1.27	6.08	32.92%

## Data Availability

The data presented in this study are available on request from the corresponding author.

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
