# Peer review of "The Size Effect of Silver Nanoparticles on Reinforcing the Mechanical Properties of Regenerated Fibers"

_molecules, 2023, doi:10.3390/molecules28041750_

Round 1

Reviewer 1 Report

The reviewer has several comments:

1, the NP size on the strength of fibers has been investigated previously, so what is the novelty of the Ag NPs?

2, it makes sense that NP may change the characters of the fibers,  so what is the reason why NP can enhance the strength?

3, Why choose 2, 20, 40, 60? with a large gap between 2 and 20?

4, Figure3, size seems not equal?

5, it should be better to detect NPs in the fibers in the SEM photos.

Author Response

Dear Editor and Reviewers, 

We appreciate the time and effort that you and the reviewers dedicated to providing feedback on our manuscript and are grateful for the insightful comments on our manuscript. We have carefully reviewed the comments and have revised the manuscript accordingly. Our response is given in the following attachment file. Thanks.

Reviewer 2 Report

The authors analyze the effect of Ag nanoparticles on the mechanical properties and structure of the regenerated silk fibers. The presentation is well arranged and easy to follow. However, some points should be explained and improved.

 - ll. 62, 63, 91, 92, 102, 103, 108: introducing two decimal figures does not correspond to reality, for instance to a deviation in setting diameters of the particles

- Figure 1 (lower part) + ll. 57-63: the results should be presented in the section Results

- l. 84: notation 2, 20, 40, 60 is rather misleading and 2.8, 19.1, 37.5, and 58.1 should be used instead. Presentation of the histograms would be useful.

- l. 100: Is there any explanation for rather non-uniform trend in 2, 0.4, 4, and 8 wt ‰? Is there any connection with rather wide distribution of the smallest NP?

- Figure 4: Better choice of axes ranges could contribute to better legibility and a comparison of a, b, c, d. For instance to unify a, b to 0-160 (strain), 0-120 (c, d), to round off the ordinate maxima.

- Table 1: too high accuracy, the authors should present an inaccuracy of the devices used indicated by the producers.

- l. 128: What is a diameter of the needle?

- Figure 5: Would it be possible to introduce mean values and standard deviations in Figure 5? (Compare with a diameter in Figure 6b.)

- l. 204: no producers introduced

Just for improvement:

- ll. 27 and 66: us – redundant

- Figure 1: to use a right ordinate (Crystallite size [nm])

- l. 82: Figure 3

- Figure 4: MPa

- l. 136: missing a word?

- Figure 9: instead of %

Author Response

(The authors gave the same response as above.)

Round 2

Reviewer 1 Report

NO

Author Response

Dear Editor and Reviewers, 

We appreciate the time and effort that you and the reviewers dedicated to providing feedback on our manuscript and are grateful for the insightful comments on our manuscript. Thanks again.

Reviewer 2 Report

The authors addressed a majority of the comments raised. However, some comments are still open.

 ad 1) Table 1: The standard deviations are not moderate, for instance for 8 (breaking strain) it attains about 30 %. In such case it is not possible to introduce the data up to 2 decimal figures (as it is completely changed with any other measurement). This also concerns the numbers introduced in the text.

 ad 3) Presentation of the histograms would be useful.

 ad 4) Is your answer projected into the manuscript?

Author Response

Dear Editor and Reviewers,

We appreciate the time and effort that you and the reviewers dedicated to providing feedback on our manuscript and are grateful for the insightful comments on our manuscript. Our response is given in the following attachment file. Thanks again.
